# Evaluation of Density-Based Spatial Clustering for Identifying Genomic Loci Associated with Ischemic Stroke in Genome-Wide Data

**DOI:** 10.3390/ijms242015355

**Published:** 2023-10-19

**Authors:** Gennady V. Khvorykh, Nikita A. Sapozhnikov, Svetlana A. Limborska, Andrey V. Khrunin

**Affiliations:** National Research Centre “Kurchatov Institute”, Kurchatov Sq. 2, Moscow 123182, Russia; gennady.khvorykh@gmail.com (G.V.K.); nikita.sapozhnikov1@gmail.com (N.A.S.); limbor.img@yandex.ru (S.A.L.)

**Keywords:** ischemic stroke, clustering algorithms, GWAS, multi-locus approach

## Abstract

The genetic architecture of ischemic stroke (IS), which is one of the leading causes of death worldwide, is complex and underexplored. The traditional approach for associative gene mapping is genome-wide association studies (GWASs), testing individual single-nucleotide polymorphisms (SNPs) across the genomes of case and control groups. The purpose of this research is to develop an alternative approach in which groups of SNPs are examined rather than individual ones. We proposed, validated and applied to real data a new workflow consisting of three key stages: grouping SNPs in clusters, inferring the haplotypes in the clusters and testing haplotypes for the association with phenotype. To group SNPs, we applied the clustering algorithms DBSCAN and HDBSCAN to linkage disequilibrium (LD) matrices, representing pairwise r^2^ values between all genotyped SNPs. These clustering algorithms have never before been applied to genotype data as part of the workflow of associative studies. In total, 883,908 SNPs and insertion/deletion polymorphisms from people of European ancestry (4929 cases and 652 controls) were processed. The subsequent testing for frequencies of haplotypes restored in the clusters of SNPs revealed dozens of genes associated with IS and suggested the complex role that protocadherin molecules play in IS. The developed workflow was validated with the use of a simulated dataset of similar ancestry and the same sample sizes. The results of classic GWASs are also provided and discussed. The considered clustering algorithms can be applied to genotypic data to identify the genomic loci associated with different qualitative traits, using the workflow presented in this research.

## 1. Introduction

Ischemic stroke (IS) is a medical condition in which the blood supply to a part of the brain is severely diminished or blocked, leading to cell death and subsequent dysfunction of the brain tissues in that area. Strokes of different types are the second most frequent causes of death worldwide [1]. IS comprises about 80% of all stroke events. The condition is multifactorial, in which the genetic component can account for up to 40% of the risk of the disease [2].

The genetics of IS, as well as many other pathological conditions and disorders, has been mainly studied with genome-wide association studies (GWASs) [3]. To date, 50 GWASs on IS have been carried out, and about 30 genomic loci have been found to be associated reproducibly with this disease [4]. However, their clinical applicability remains questionable as only a small part of the phenotypic variability has been explained [5], highlighting the under-exploration of IS and suggesting potential for the discovery of new associations (genes) in the future. A possible solution for this is to process available datasets using new approaches.

The data from GWASs consist of hundreds of thousands of single-nucleotide polymorphisms (SNPs) distributed throughout the genomes of individuals both with (cases) and without disease (controls). Traditionally, each SNP is considered to be independent and tested statistically for its association with disease separately. However, the power of statistical testing might not be enough to take into consideration the associations between genetic variants with small effects and phenotypes. Additionally, the traditional approach does not account for SNP-to-SNP interactions existing because of linkage disequilibrium (LD). These limiting capabilities of classic GWASs can be avoided with multi-loci approaches such as testing sets of SNPs instead of individual polymorphisms.

There are several multi-loci approaches in associative studies, including haplotype-based and SNP-set analyses, e.g., gene-based and pathway-based analyses [6]. Haplotype-based analysis accounts for LD between the SNPs, while SNP sets take into consideration all SNPs within a gene or genes from a particular pathway to estimate the joint effects of allelic variants on the trait [7,8,9,10]. However, the efficacy of SNP-set-based approaches can substantially depend on the prior biological information about the analyzed components. Additionally, the degree of LD between the loci can also be important due to its inequality across parts of the genome. In contrast, haplotypes do not require any prior knowledge and are more appropriate in the context of the block-like structure of common human genetic variation [11].

Although the original data in GWASs are provided as individual genotypes, the haplotypes, which are the sequences of allelic variants along a single chromosome inherited together, can be inferred statistically from genotypes [12]. However, in this case, one needs to decide how to split the SNPs into blocks, in which the haplotypes will be further inferred and tested. This task is sometimes referred to as the problem of haplotype block partitioning. It can be solved, for example, by minimizing the number of SNPs that uniquely identify the common haplotypes in each block or by finding sets of SNPs with a high pairwise LD that are usually called haplotype blocks or LD-blocks [13].

In the last 20 years, several approaches to building LD-blocks have been proposed. The polymorphisms can be arranged either consecutively or not within the block. In the first case, many algorithms were described. Among them were the estimation of confidence intervals for pairwise D’ values [11], the four-gamete test for checking the recombination crossover between the loci tested [14], the Solid Spine algorithm, defining the block as a set of loci in high LD with the first and last loci [15], the Markov chain Monte Carlo algorithm to identify blocks of linkage disequilibrium [16], etc. The second group of algorithms was less representative. For example, the PCA method can be applied, resulting in LD-correlated groups of SNPs that are not a contiguous DNA fragment [17]. In this research, we aim to enlarge the methods of genome partitioning into LD groups where SNPs are not necessarily adjacent. From a general point of view, this task can be undertaken with clustering algorithms using the methods of unsupervised machine learning. The key parameter in clustering is the similarity of objects within the group.

LD between two SNPs is usually expressed with D’ or r2 metrics that can be considered as a measure of the similarity of corresponding SNPs. Therefore, the symmetric matrix having pairwise LD measures can be naturally processed with unsupervised methods of machine learning. The family of such methods is quite large. However, only a few of the methods have been applied so far in genomic association studies. For example, hierarchical clustering has permitted identification of SNPs associated with leaf sodium accumulation in A. thaliana, and these SNPs in high LD were not necessarily contiguous [18]. Affinity propagation allowed for clustering SNPs into blocks, which helped researchers to choose the kernel sizes while applying convolutional neural networks to soybean data to predict quantitative phenotypes [19].

It is easy to note that the LD matrix becomes an adjacency matrix, representing an undirected graph with SNP vertices, after a given threshold value of the LD measure is applied to the LD matrix. This allows for the utilization of graph partitioning methods, especially those defining sets of vertices, as in the case of the clique problem. The application of the Bron–Kerbosch algorithm to find maximal cliques in the graph obtained from the LD matrix resulted in clusters of strongly correlated SNPs that were not necessarily physically consecutive and agreed better than existing methods of LD-block partitioning with the recombination hotspot locations determined by sperm-typing experiments [20].

Continuing to explore unsupervised methods for the genome partition as a part of the pipeline to discover associations between the genetic variants and common diseases, we investigated the density-based spatial clustering algorithms DBSCAN [21] and HDBSCAN [22]. These methods define a cluster as a dense component able to grow in any direction in which the density leads. This feature means that the algorithms can identify clusters of an arbitrary shape and the dataset can have noise and outliers. The methods do not require the specification of the number of clusters in advance and possess good scalability.

Previously, DBSCAN was applied to cluster SNPs. It was a part of the hybrid approach to defining latent variables of a Bayesian network to discover the genetic bases of complex phenotypes. As a result, the GWAS approach utilizing DBSCAN to model LD outperformed the approach based on traditional LD modeling through blocks of contiguous SNPs [23]. This investigation forced us to consider density-based spatial algorithms as promising for the purpose of our research. We applied DBSCAN and HDBSCAN algorithms to simulated and real datasets of close ancestry and similar sample sizes.

## 2. Results

### 2.1. Synthetic Data

#### 2.1.1. Data Simulation

To validate the cluster-based approach, we simulated the genotype–phenotype dataset consisting of 32,028 SNPs from chromosome 22 and represented by the genotypes of 4929 cases and 652 controls. In order to ensure the synthetic data were realistic, we applied the haplotypes of individuals from CEU populations of the 1000 Genomes project as the reference data. The distributions of minor allele frequencies (MAFs) for reference and simulated data were found to be close (Appendix A). Since the cluster-based approach exploits the LD, we also compared the decay of LD in both the reference and simulated datasets. The simulation was found to influence the LD. It decreased faster by distance in the simulated dataset as compared to the reference ones (Appendix A). Although the differences were noticed on distances higher than 50 Kbp, the length of clusters in the real data can be on the same scale (see Appendix A). Breaking LD during the simulation was a challenge in obtaining the dataset for validation of the cluster-based approach. In any event, the synthetic dataset with five disease SNPs linked to other SNPs was created and analyzed with classic and cluster-based GWAS approaches.

#### 2.1.2. Associative Studies

The histograms of *p*-values obtained with statistical tests and logistic regression showed almost uniform distributions (Appendix A). All five disease SNPs demonstrated unadjusted *p*-values less than 5 × 10^−8^ (Appendix A) and expected odds ratios (ORs) close to 1.5 (Appendix A).

The evaluation of clusterization parameters revealed the optimal ones to be ‘eps’ = 0.35 and ‘min_samples’ = 2 for DBSCAN and ‘cluster_selection_epsilon’ = 0.25 and ‘min_cluster_size’ = 2 for HDBSCAN. The histograms of unadjusted *p*-values obtained by the haplotype tests applied to blocks corresponding to the clusters identified with DBSCAN and HDBSCAN algorithms are provided in Appendix A. Five clusters were found to be associated significantly with the phenotype at the threshold *p*-value of 5 × 10^−8^. They corresponded exactly to five disease SNPs. The heatmaps of LD within these clusters, as well as the SNPs forming them, are shown in Appendix A. All five disease SNPs introduced upon the simulation of genotype–phenotype data were identified with a cluster-based workflow, thus validating this approach.

### 2.2. Real Genome-Wide Data of Ischemic Stroke

#### 2.2.1. Traditional GWAS

Five Manhattan plots are shown in Appendix A, representing the results of the traditional GWAS approach realized via five evaluations. The lists of SNPs associated significantly with IS for each testing are provided in Appendix A. Non-zero intersections of these lists are shown in Figure 1. The union of these lists had 29 SNPs. They were compared with those found in previous GWASs.

There were 92 of the 353 SNPs downloaded from GWAS Central represented in the whole list of SNPs available in our research. The intersection of 29 significant SNPs identified in this research as being associated with IS with SNPs downloaded from GWAS Central did not reveal common SNPs. The minimum asymptotic *p*-value for the 92 SNPs from GWAS Central obtained in our research equaled 3.1 × 10^−3^. The results of the annotation of significant SNPs in terms of genes and SO are provided in Appendix A. The frequency of SO terms for all SNPs available under this research and the significant ones are provided in Table 1.

The majority of significant SNPs were intergenic. Since they can either affect upstream and/or downstream genes directly or be in LD with the SNPs that affect the functions of these genes, both examples nearest to such SNP genes were considered. Under this assumption, the annotation of 29 selected SNPs resulted in 35 genes and loci (see Appendix A), 27 of which were recognized by the DAVID web service. However, the tests for over-representation of these genes in DAVID databases did not reveal significant results (FDR < 0.05).

The comparison of 35 genes found in this research with the genes downloaded from GWAS Central, DisGeNET, and Monarch Initiative projects revealed one common gene, *RUNX1,* in each pairwise intersection, and the intersection with IDG genes resulted in one common gene, *GPR26*. The gene *RUNX1* was annotated to intron variant rs926202 (Appendix A), significantly associated with IS with allelic (*p*-value = 0.002), genotypic (*p*-value = 0.012) and trend (*p*-value = 0.003) tests. *RUNX1* encodes runt-related transcription factor 1, a protein that regulates the differentiation of hematopoietic stem cells into mature blood cells [24].

Another significant intron variant—SNP rs34201757 (*p*-value = 0.01, allelic and trend tests)—was in gene *GPR26* (Appendix A). This gene encodes the G-protein coupled receptor protein belonging to a large family of membrane proteins involved in cellular responses to environmental stimuli, neurotransmitters, and hormones. To our knowledge, *GPR26* had not been associated with stroke in previous research.

#### 2.2.2. Cluster-Based Approach

The evaluation of clusterization parameters resulted in ‘eps’ and ‘cluster_selection_epsilon’ being equal to 0.41 and ‘min_samples’ and ‘min_cluster_size’ being equal to 5. These values occurred frequently among the best values obtained for different chromosomes by DBSCAN and HDBSCAN. The changes in the metrics of clusterization quality dependent on changes in clusterization parameters for chromosome 1 are shown in Figure 2.

The optimal parameters of clusterization were applied to group SNPs by their pairwise LD into clusters. The descriptive statistics of clusters thus retrieved and other objects analyzed are provided in Table 2.

As expected, HDBSAN was more effective than DBSCAN, resulting in a lower number of polymorphisms unassigned to any cluster. HDBSCAN produced 1.6% more clusters than DBSCAN. The results of clusterization were also described on the level of chromosomes. The number of clusters produced by the two algorithms for each autosome is shown in Figure 3. The median of the differences between the numbers of clusters obtained with HDBSCAN and DBSCAN for each autosome was 1.5%.

The histograms of cluster sizes for DBSCAN and HDBSCAN are provided in Figure 4. Both had right-skewed distributions with long tails. According to the Mann–Whitney U test, these distributions are different (*p*-value < 2.2 × 10^−16^). DBSCAN showed the highest number of clusters sized five (N = 10,354), while HDBSCAN demonstrated the most frequent cluster sized six (N = 5515). DBSCAN formed more clusters sized lesser than or equal to seven than HDBSCAN, while HDBSCAN formed more clusters sized greater than or equal to eight than DBSCAN.

In order to compare the number of clusters, the Silhouette coefficient and the Calinski and Harabasz score between the chromosomes, we normalized the corresponding values by the number of polymorphisms at each chromosome and calculated the z-scores. The absolute values of the z-score for all three metrics were within three standard deviations. The positive and negative values of the z-scores were equally observed (Appendix A). As seen, the distribution of metrics considered did not reveal statistically significant differences between the chromosomes; thus, we may conclude that chromosomes, or their parts as in the case of chromosomes 1, 2, and 6, behaved similarly under clusterization.

The clusters formed by DBSCAN and HDBSCAN demonstrated a mosaic-like structure in the sense that the polymorphisms from a particular cluster were not necessarily neighbors on the genome (Figure 5).

The haplotype tests resulted in 68 and 79 LD-blocks associated significantly with IS for DBSCAN and HDBSCAN, respectively (see Appendix A). The distributions of the sizes of such blocks were similar for the two clustering algorithms. The *p*-value of the Mann–Whitney U test was equal to 0.11, and the median sizes were seven and eight for DBSCAN and HDBSCAN, respectively (Figure 6).

The selected LD-blocks were built of 666 (DBSCAN) and 892 (HDBSCAN) polymorphisms, almost half of which were common to both algorithms (Figure 7). Neither DBSCAN nor HDBSCAN showed common polymorphisms with GWAS Central.

Defining the genomic context of these LD-blocks, we excluded intergenic polymorphisms for simplicity of interpretation and received 98 and 122 genes associated with IS for DBSCAN and HDBSCAN, respectively. The intersection of these genes resulted in 88 common genes (Appendix A). The comparison of these genes with those downloaded from online projects revealed some common genes. They are listed in Table 3.

Most of the genes identified in our research using a cluster-based approach (N = 88) were recognized by MSigDB (N = 71) and tested for over-representation in the gene collections of this project. Sixteen genes were found to belong to the protocadherin gamma subfamilies A and B. These proteins are involved in the biological processes of adhesion to the plasma membrane [25].

#### 2.2.3. Classical GWAS vs. Cluster-Based Approach

The classic GWAS and cluster-based approach revealed four SNPs significantly associated with IS in common (Figure 7). Three of them were identified with both DBSCAN and HDBSCAN algorithms (rs6754311, rs13072547, rs12592594) and one—rs11987006—was identified only with DBSCAN. The polymorphisms rs6754311 and rs13072547 lay inside introns of *DARS* and *ERC2* genes, respectively. rs12592594 was annotated as ‘intragenic_variant’ of gene *SEMA6D*. The polymorphism rs11987006 lay between gene *RP11-465K16.1* and pseudogene *RNU6-356P*.

The intersection of the gene lists obtained with the classic GWAS (N = 35) and the cluster-based approach (N = 88) showed seven genes in common. Three of them (*DARS*, *ERC2*, and *SEMA6D*) were supported by common SNPs and the other four (*MAP3K19*, *MAP3K4*, *FAM107B*, and *RUNX1*) did not possess common SNPs from the list of SNPs obtained with the classic GWAS.

## 3. Discussion

### 3.1. Traditional GWAS

Through performing a traditional GWAS, we identified 29 SNPs associated significantly with IS. They were obtained by five tests that corresponded to different models of inheritance (genotypic, dominant, and recessive) and types of tests (the Cochran–Armitage test for trend and the allele frequency test). The minimum number of SNPs was revealed via the recessive model (N = 4) and the maximum number of SNPs was revealed via the allelic test (N = 18). As seen in Figure 1, the lists of SNPs obtained using different models were almost unique. The power of the tests considered depends on many factors (sample size, inheritance model, MAFs, Hardy–Weinberg equilibrium, etc.), which makes it difficult to interpret the differences in the numbers of significant SNPs obtained [26]. Nevertheless, when the particular inheritance model for most of the loci is not known, the results of all modeling can be suggestive and considerable. Therefore, the lists of significant SNPs obtained by five tests are worth considering.

Although the 92 SNPs from GWAS Central that are known to be associated with IS were available in our research, we did not identify them as being significantly associated with IS. On the one hand, inconsistency with previous studies was expected. Assuming that the 29 SNPs were selected by chance, the probability that at least one significant SNP identified by us was from GWAS Central would be 1 − $C_{\left(N-29\right)}^{29}/C_{N}^{29}$ = 3.0 × 10^−3^, where N = 883,908. On the other hand, the SNPs were not selected randomly. We considered them to be risk factors for disease, meaning that certain biological processes underlay them. Here, we encountered the problem of the reproducibility of the GWAS. The comparison of SNPs was not only identified in different studies but also considering those that are in LD with them could be of help. The comparison of genes was another option.

The SNPs available in this research allowed us to test almost all genes known to be associated with IS from previous studies (Table 4). The annotation of 29 SNPs that we detected resulted in 35 genes. We observed only one intersection of these genes with each of three online resources (GWAS Central, DisGeNET, and Monarch Initiative)—gene *RUNX1*. Its association with IS was identified previously with a classic GWAS on individuals of African ancestry and Europeans [27]. However, the role of *RUNX1* in IS has not yet been established. Its product is a key transcriptional regulator of the differentiation of hematopoietic stem cells into mature blood cells [24]. The protein RUNX1 is crucial in hematopoiesis during embryonic development but its role in adults awaits clarification [28]. Recently, it was associated with cardiac remodeling after myocardial infarction (MI). Clinically, the remodeling is manifested as left ventricular wall thinning, dilation and reduced contractility and was found to be accompanied by elevation in *RUNX1* expression in cardiomyocytes. Reduction in *Runx1* function by knockout preserved cardiac contractility and prevented adverse cardiac remodeling in mice due to its effect on cardiac sarcoplasmic reticulum calcium uptake [29]. Up-regulation of *RUNX1* in the MI heart may be in response to different reasons, including changes in electrical activity, cell–cell communication or as a result of physical strain or tension [30]. It can also be triggered by inflammatory stimuli from the infarcted tissue [31]. Inflammation is one of the important factors in IS risk [32]. One can propose that the association of *RUNX1* with IS is also due to its involvement in vascular remodeling in response to inflammation. It may potentially occur through the ability of RUNX1 to modulate the inflammatory phenotype of the cells. One example of this is through the further stimulation of inflammation via enhancing the production of inflammatory cytokines in immune cells, such as IL-1β and IL-6, due to its interaction with the p50 subunit of NF-κB [33]. This is known to be a major orchestrator of the inflammatory actions of cytokines in cardiovascular and metabolic diseases [34].

Another gene that was identified using the classic GWAS and mentioned in an external resource, i.e., IDG, was *GPR26*. The gene encodes the G-protein coupled receptor (GPCR) that belongs to a well-established druggable family of proteins. GPCR genes are expressed in the brain and their products mediate the signal of transduction in the central nervous system. Their roles in the development of anxiety and mood disorders as well as in the mechanism of antidepressant therapies are widely studied [35]. GRP26 deficiency was associated with the susceptibility to obesity and diabetes [36,37]. Targeted deletion of *Gpr26* in rodents caused glucose intolerance, hyperinsulinemia and dyslipidemia [36]. In patients with type 2 diabetes *GPR26* was downregulated due to chronic hyperglycemic conditions and accompanied by increased production of reactive oxygen species, pro-inflammatory monocyte activation and apoptosis [37]. Both obesity and diabetes are known risk factors for IS due to cardiovascular complications. However, the association of *GPR26* with IS itself was demonstrated for the first time. This might shed light on common genetic determinants of different mechanisms of pathogenesis of the disease and attract attention to GPR26 as the therapeutic target under IS.

### 3.2. GWAS Based on Clustering

We reanalyzed the real data from the GWAS of IS by comparing the frequencies of haplotypes inferred statistically in LD-blocks for the case and control groups to localize the genetic variations responsible for the disease. The novelty of the proposed workflow lay in how the LD-blocks for inferring the haplotypes were defined. To partition the genome into groups of correlated SNPs, we applied the density-based clustering algorithms DBSCAN and HDBSCAN. Their use provided a number of advantages over previously used methods.

First, previous approaches considered SNPs as being contiguous within the genome sequence and corresponding to non-overlapping LD-blocks; however, our approach did not stipulate this. This resulted in LD-blocks that were mosaic in the space of the genome regions tested, better reflecting the complex structure of LD patterns. The methodology allowed interactions to be accounted for between SNPs, including not only neighboring markers but also long-distance ones (Appendix A). Being non-linear, such a model better fits the polygenic nature of complex diseases.

Second, the clustering algorithms DBSCAN and HDBSCAN created clusters of polymorphisms without the number of clusters being known in advance. This allowed discovering any number of clusters in data that are essential in solving the problem of genome partition. However, there are other parameters of algorithms applied that affect the results of clusterization. Since there is no formula for assigning them, we evaluated these parameters experimentally by maximizing the metrics of the quality of clusterization (Silhouette coefficient and Calinski and Harabasz score). The parameters thus obtained were quite reasonable.

The highest number of clusters formed by the two algorithms were sized five or higher. That agrees with ‘min_samples’ and ‘min_cluster_size’ equal to 5. In addition, the medians of the LD-blocks associated with IS were seven and eight for DBSCAN and HDBSCAN, respectively. This result also supports the chosen values of the parameters. However, the estimations obtained may be sensitive to the number of polymorphisms measured and the patterns of LD. The other genotypic data will probably require their own evaluation of the parameters of clusterization.

‘eps’ and ‘cluster_selection_epsilon’ equal to 0.41 corresponded to r^2^ for polymorphisms in the cluster equal to 0.59 that met our expectation for generating groups of correlated polymorphisms. We believe the proposed evaluation of clusterization parameters diminishes the possible bias because of the choice of these parameters.

The clustering algorithms studied were compared and applied to the same dataset. HDBSCAN grouped more polymorphisms into clusters than DBSCAN, thus leaving fewer polymorphisms as noise. The efficiency values of HDBSCAN and DBSCAN were 0.69 and 0.55, respectively. The more polymorphisms clustered, the more genomic regions that could be tested for association with disease. From this point of view, HDBSCAN was more efficient than DBSCAN.

Interesting observations were made while comparing the distributions of cluster sizes. First, HDBSCAN formed more clusters of sizes greater than or equal to eight than DBSCAN. This observation agrees with the peculiarity of HDBSCAN. Unlike DBSCAN, it identifies clusters of any density. Second, the distributions of sizes of the LD-blocks selected were similar. Intuitively, the larger the LD-block, the higher the probability of the recombination breaking the long haplotypes. Therefore, less chance that frequencies of corresponding haplotypes will be significantly different in the case and control groups of people.

DBSCAN and HDBSCAN are intrinsically connected. They are built on the same principles. These two clustering algorithms produced a slightly different number of clusters, which resulted in a different number of LD-blocks, polymorphisms and finally risk genes. The transition from LD-blocks to risk genes was simplified. It involved the annotation of the individual polymorphisms making up an LD-block. The annotation was based on the intersection of the genomic coordinates of the polymorphisms and genes. Strictly speaking, two things should be considered here. First, the probability of change in gene functional activity by a non-synonymous single-nucleotide polymorphism should be evaluated. Second, such an evaluation should be performed for haplotypes rather than for individual polymorphisms from LD-blocks. To our knowledge, the estimation of the cooperative effect on gene functionality of at least two SNPs is a complicated task, not to mention the effect of several SNPs. That is why, in this research, we were satisfied with the results of the annotation of individual SNPs from LD-blocks in terms of SO. However, the lists of genes thus obtained after DBSCAN and HDBSCAN were questionable. Therefore, we decided to intersect them rather than to contrast them. Finally, 88 genes common to both algorithms were selected and analyzed. They represented 66.7% of all the risk genes identified using both clustering algorithms.

The workflow proposed has some limitations. First, one needs to be aware of LD patterns since one of the key steps is the clusterization of SNPs. No LD means no clusters, and no clusters means no loci associated with the phenotype. It is possible to generate the synthetic data with risk SNPs not linked to any other SNPs. They would be missed using our approach, since DBSCAN and HDBSCAN will eliminate them as noise. The question is how often causative SNPs without linkage occur in real-life data. If there is a concern about such a situation, it can be easily overcome by testing individual SNPs with a chi-square test or by logistic regression. In other words, the cluster-based approach and classic GWAS complement each other, and that is why we applied both of them in our research. It is clear that the joint application of the single-locus and multiple-loci approaches requires further investigation. Second, the annotation of the results of the haplotype test in terms of genes is questionable. As mentioned above, we simplified it by using individual SNPs instead of groups of SNPs from the LD-block associated with the phenotype, since it was not clear how to estimate the cooperative effect of SNPs on genes. In addition, we considered only the results of direct gene annotation, since ‘intergenic’ ones were difficult to interpret because of the mosaic structure of the clusters obtained.

Testing selected genes for over-representation in gene set databases suggested the possible role of protocadherins (PCDHs) in IS. PCDHs are a type of cell adhesion molecule predominantly expressed in the central nervous system (CNS). Being crucial for CNS development, including neurite initiation and outgrowth, axon pathway finding and fasciculation, synapse formation and stabilization as well as neuron survival, several PCDH genes have been associated with neurodevelopmental disorders [38,39]. Due to interaction with the Wnt signaling pathway, PCDHs have also been proposed as affecting the development of Alzheimer’s disease and neuronal survival in focal ischemic injury [40]. The role of PCDHs in ischemic stroke has not yet been explored. Only one gene—*PCDH7*—was associated with the incidence of stroke [41]. More data were obtained in studies of myocardium functions under ischemic cardiomyopathy and ischemic heart diseases where expression of PCDHs, including protocadherin gamma family members, was also correlated with stroke volume, ventricular dysfunction and the benefit of cardiac surgery [42,43]. Overexpression of several such genes was also found in patients with myocardial infarction [44]. These results demonstrated the association of PCDH genes with cardiac dysfunction and suggested that their expression level could be a diagnostic marker of the dysfunction progress. Based on the results of our study, we can hypothesize that the complex dysfunction of cell adhesion-related processes in the cardiovascular system predates an IS event. The results of a recent epigenome-wide study suggested that they might be correlated with atherosclerosis [45], particularly due to the roles of PCDHs in cell-to-cell interactions between smooth vascular muscle cells in the aortic wall and atherosclerotic plaque progression and/or destabilization [44,46].

The presented steps of data processing form a pipeline that provides an alternative to the steps applied routinely in GWASs. Following Zitong Li et al. [18], we can refer to it as a cluster-based GWAS. The apparent difference between the two approaches is the choice of the tested marker. The classic GWAS tests individual polymorphisms, while the cluster-based GWAS tests groups of correlated polymorphisms. The first method is often referred to as single-locus, while the second one is referred to as a multi-loci approach. We will devote some of the text to comparing them.

The most intriguing observation in our study is that the classic and cluster-based GWASs had only four SNPs in common. On the one hand, we may assume that single SNP testing does not allow for accounting for portions of the total allele frequency associated with particular haplotypes whose differences between patients (case) and control may be really determinative. On the other hand, it seems to be correlated with the threshold of association significance specified in each analysis. The cluster-based approach operated with associations that are significant at an unadjusted *p*-value ≤ 1 × 10^−6^ that is frequently considered as “suggestive” in the results of traditional GWASs and that can be proposed as more realistic and plausibly related to the biology of the traits [47]. Furthermore, neither the classic nor the cluster-based GWAS possessed common SNPs from GWAS Central, thus emphasizing once more the difficulties in comparing GWAS results from different sources on the level of SNPs. At the same time, both approaches and GWAS Central identified the common gene *RUNX1* (Table 3). How is that possible? The classic GWAS indicated this gene with intron variant rs926202 (chr21:37128336). GWAS Central provided *RUNX1* accompanied by intron variant rs116262092 (chr21:36442465). Both DBSCAN and HDBSCAN led to the same LD-block (chr21:36897326-37049608) composed of 16 SNPs (Appendix A), each of which can potentially affect the function of *RUNX1*. This LD-block lies inside *RUNX1*, covering 152,283 out of 1,216,868 bp of its length (Appendix A).

This research belongs to the category of associative studies, a large family of investigations aimed at the discovery of the genetic bases of quantitative and qualitative traits. In our case, the trait is the multifactorial disease, and the potential translation of the study to clinical practice is worth mentioning. Being an associative study, the research has all the limits and merits of this type of investigation. To name a few limitations, the SNPs and genes found to be associated with a phenotype are only the candidate ones. We do not know whether they are or are not the cause of the disease. The cause of the disease can be established only after all the components involved in the development of the disease are established and their role in it is described. This requires appropriate investigations of specific designs. Another problem with the translation of the results of associative studies into clinical practice is the low reproducibility of the associations (risk loci) identified. There is also a bioinformatics challenge in building prediction models to estimate the risk of the disease once the genetic variants are provided.

We believe that our research, using a new workflow for associative studies, will have the ability to reduce the gap between associative studies and clinical practice through presenting a valuable tool and pointing out new genes involved in the pathogenesis of ischemic stroke. In addition to associative studies, the density-based spatial clustering algorithms DBSCAN and HDBSCAN can be applied for solving other current challenges in genomics, such as finding tag SNPs, detecting outliers while testing the genotypes within a certain population for homogeneity and reducing the dimensions of the genotypic data for the supervised machine learning algorithms to be applied.

## 4. Materials and Methods

The creation of input files for all programs used in this research, the analysis of output files created as well as the automation of all computational steps were performed with custom scripts in Perl v5.30.0 [48], Bash 5.0.17 [49], Python 3.8.3 [50] and R 4.2.1 [51]. The description of the essential stages of the workflow realized is set out below.

### 4.1. Real Data Preprocessing

The individual genotypic and phenotypic data were downloaded from the international database of genotypes and phenotypes, dbGaP, under accession number phs000615.v1.p1. According to the project description, the data were obtained from a large study of about 6000 patients with ischemic stroke and about the same number of people without the disease conducted by 13 genetic centers in the US and 11 in Europe [52]. From them, individuals aged over 55 years who identified themselves as white were selected by us.

Only the polymorphisms located on autosomes were considered in the study. To check the population structure, the genotypic data were processed with the ADMIXTURE program [53] and the PCA method implemented in the Plink 1.9 program [54]. ADMIXTURE showed the presence of individuals with different ancestral components in cases and controls. A two-component model was applied and 1035 people with a substantial contribution from the second ancestral component (i.e., greater than 0.046) were removed from the dataset. Twenty individuals with eigenvector values greater than six standard deviations from the mean were additionally removed according to the results of the PCA analysis of the first twenty principal components. The final numbers of individuals were 4929 and 652 in the case and control groups, respectively.

The quality of genotyping was evaluated using the Plink 1.9 program. Polymorphisms and individuals with a proportion of missing genotypes of more than 20% were filtered out. The genotypes of people from the control group were tested for concordance with the Hardy–Weinberg law using the Plink 1.9 program. The polymorphisms with *p*-value < 1 × 10^−5^ were excluded from further consideration. The total number of polymorphisms left equaled 883,908, of which 883,749 were SNPs and 159 short insertion/deletions. The missing genotypes were restored statistically using the LinkImpute program [55]. The original genotypic data were under human genome assembly GRCh37, which was kept while processing the data.

### 4.2. Traditional GWAS

The traditional GWAS was implemented via the chi-square test for independence, Fisher’s exact test and the Cochran–Armitage test for trend. In addition to the genotypic (additive) model, the dominant and recessive models of inheritance were also applied. Additionally, the allele frequency distribution was also assessed. All five tests were constructed with the Plink 1.9 tool using the argument ‘--model’. The simulated dataset was also tested with logistic regression using the argument ‘--logistic’ of the Plink tool. In the case of the real dataset, the resulting asymptotic *p*-values were corrected for multiple testing with the *p*.adjust() R function using the Bonferroni method, and the significance threshold was set at 0.05. In the case of the synthetic dataset, the SNPs associated significantly with the phenotype were selected by the unadjusted *p*-value threshold of 5 × 10^−8^.

### 4.3. Cluster Approach

The real genotypic data were split into 22 datasets possessing the polymorphisms from the corresponding autosomes. Each dataset was transformed into an LD matrix with Plink 1.9 using the argument ‘--r2 square’. Thus, the text files obtained contained pairwise r^2^ measures of LD between all polymorphisms of the corresponding chromosome. The datasets were further transformed into 1–r^2^ matrices and clusterized with DBSCAN using the Python scikit-learn 1.1.3 package [56] and Python module HDBSCAN [57].

The clustering algorithms DBSCAN and HDBSCAN have some parameters that can be modified. The following five parameters were considered to essentially influence the clusters formed: ‘eps’ and ‘min_samples’ for DBSCAN and ‘cluster_selection_epsilon’, ‘min_samples’ and ‘min_cluster_size’ for HDBSCAN. ‘eps’ defines the maximum distance between two neighboring points, ‘min_samples’ sets the number of samples (the loci genotyped) in a neighborhood for a point (locus) to be considered a core point, ‘cluster_selection_epsilon’ sets a distance threshold to merge the clusters below this value and ‘min_cluster_size’ sets the minimum size of clusters. We evaluated these five parameters computationally by maximizing the Silhouette coefficient and the Calinski and Harabasz score that were calculated with the functions silhouette_score() and calinski_harabasz_score() from the scikit-learn package, respectively. The Silhouette coefficient shows how far clusters are apart from each other, and the Calinski and Harabasz score measures how similar objects are in a cluster, compared to other clusters.

The parameters were separately evaluated for each chromosome by DBSCAN and HDBSCAN. First, we fixed the ‘eps’ and ‘cluster_selection_epsilon’ to 0.5 and varied ‘min_samples’ and ‘min_cluster_size’ from 2 to 10 in increments of 1. In the case of HDBSCAN, ‘min_samples’ and ‘min_cluster_size’ were always the same. Once the best value of ‘min_samples’ and ‘min_cluster_size’ was determined, it was fixed, and ‘eps’ and ‘cluster_selection_epsilon’ were varied from 0.01 to 1.00 in increments of 0.1. The best values of the five parameters discussed were chosen as the most frequent ones for all chromosomes processed with DBSCAN and HDBSCAN. The other arguments of clusterization functions were default.

Since it was discovered that the haplotypes on chromosomes 1 and 2 were not inferred from the clusters obtained within a reasonable time because of computational load, we split these chromosomes into short and long arms and processed the polymorphisms on each arm separately. Because of the complex structure of the major histocompatibility complex (MHC) located on the short arm of chromosome 6 (e.g., high levels of variation, complex linkage disequilibrium and the highest density of genes in the human genome) [58,59], we excluded the MHC region from consideration. Its borders were defined by rs2328848 and rs7748483, located at positions 25,175,316 and 33,853,071, respectively, with the overall length of the excluded region being 8,677,755 bp. The remainder of the autosomes were processed as they were found. The number of polymorphisms left for clusterization equaled 871,625.

We defined the haplotypes as the combinations of alleles of polymorphisms from a particular cluster (LD-block) of the chromosome. The haplotypes were inferred with the Plink 1.7 program [60] using the ‘--hap-assoc’ argument. We considered LD-blocks of sizes from two to the maximum number of polymorphisms in a cluster. Assuming that markers are on a chromosome according to their genomic coordinates rising from left to right, the first two on the furthest left of the polymorphisms were treated, then the third, fourth, etc. Thus, the number of polymorphisms processed within a cluster equaled {2, 3, …, n − 1, n}, where n is the size of the given cluster.

LD-blocks were considered as a categorical feature whose values were inferred haplotypes and were tested for association with IS using a chi-square test of independence implemented in Plink 1.7 as an omnibus test. For each cluster, we chose one LD-block with the minimal asymptotic *p*-value. If several LD-blocks met this condition within a cluster, the largest by size was selected. In the case of the real dataset, the *p*-values of selected LD-blocks were subjected to correction for multiple comparisons with the Bonferroni method using the *p*.adjust() R function. The number of tests equaled the total number of LD-blocks tested, which is the total number of clusters. The LD-block was considered to be associated with the phenotype if the adjusted *p*-value was less than 0.05. In the case of the simulated dataset, the LD-blocks were selected with an unadjusted *p*-value of less than 5 × 10^−8^.

The efficiency of clusterization was defined as the decimal fraction of polymorphisms clusterized from the total number of polymorphisms available for clusterization. The size of cluster, LD-block or haplotype was the number of polymorphisms that the corresponding object involved. To compare them between the chromosomes, the *z*-score was calculated according to the formula:$$z_{i}=\frac{x_{i}-\mu }{\sigma }$$
where *μ* and *σ* are the mean and standard deviation of variable *x*, respectively.

### 4.4. Post-GWAS Analysis

The polymorphisms significantly associated with IS after the classic GWAS as well as the polymorphisms forming the LD-blocks significantly associated with IS after the cluster-based approach were annotated in terms of sequence ontology (SO) [61] and genes with the snpEff 5.1 program [62], using the enclosed database GRCh37.87. In the case of the classic GWAS, we considered all genes that resulted from the annotation for each of the five tests made. In the case of the cluster-based approach, the results of annotation having the status ‘intergenic’ were eliminated from further consideration and the lists of genes obtained with DBSCAN and HDBSCAN were intersected to obtain common entities that were further analyzed. The sets of genes thus obtained were tested for over-representation in the sets of human genes provided in the Molecular Signature Database (MSigDB) [63]. The functional annotation of lists of genes was made with DAVID [64]. The level of significance was set up as FDR < 0.05.

In order to compare the results obtained with those published earlier, the following databases available online were utilized. SNPs and genes found to be associated with IS in previous GWASs were downloaded from the GWAS Central project by keyword ‘stroke’ and −log10 (*p*-value) > 5.0 (gwascentral.org, accessed on 2 April 2023) [65]. These data contained 353 SNPs annotated by 400 genes from 25 studies. We also downloaded genes associated with IS by sending the query ‘ischemic stroke’ to the Monarch Initiative (monarchinitiative.org, accessed on 13 June 2023) [66] and DisGeNET (disgenet.org, accessed on 7 October 2022) [67] platforms.

Although the phenotype studied represents the onset of IS, we were also interested in the therapeutic potential of the genes identified, assuming the existence of biological processes common to both the onset and the course of the disease. In this sense, we considered the genes representing the most common drug-targeted proteins. Such genes were obtained from the Illuminating the Druggable Genome (IDG) program (druggablegenome.net, accessed on 21 March 2023). Since each of the four online resources mentioned above had an essential number of unique genes (Appendix A), we considered each of the lists of genes separately in the research.

We annotated all polymorphisms (N = 883,908) available in our research in terms of genes with snpEff 5.1 and compared the list of genes thus obtained with the lists of genes downloaded from online resources. The numbers of genes downloaded and tested in our research are provided in Table 4**.**

The Mann–Whitney U tests were made with the wilcox.test() R base function. The results obtained were visualized with the R packages ggplot2 [68], qqman [69], ggvenn [70], UpSetR [71] and karyoploteR [72] as well as with the custom script using the Python imaging library Pillow 9.0.1 [73].

### 4.5. Data Simulation

To validate the approach used at the discovery stage, we simulated data since we did not possess the real datasets with confirmed disease loci and of the same sample sizes. Concerning the allele frequencies and LD in synthetic data, to be realistic, among different approaches to simulate genotype–phenotype data we chose one that utilized the real data as the reference panel. The dataset of the given sample size having specified disease loci was generated with Hapgen 2.2.0 software [74]. The input data were the phased genotypes from Phase I of the 1000 Genomes project [75], the corresponding recombination map and the list of assigned disease SNPs. After preprocessing, the reference panel represented 32,028 SNPs from chromosome 22 and 99 unrelated individuals from the CEU population. The disease SNPs were selected as having MAFs between 0,2 and 0.4, separated from each other by more than 100 Kbp and lying in the regions of high LD, the linkage of SNPs with the identified risk one being preserved upon simulation. The selection of such SNPs recalls the selection of proxy or tagSNPs in associative analysis. To assist in the identification of such SNPs, we estimated the LD haplotype blocks with the Gabriel algorithm [11] implemented in Plink 1.9 (--blocks argument). Finally, the following five disease SNPs were selected and implemented for simulation: rs4823464, rs137425, rs3761422, rs9606478 and rs7292279. The ORs of disease risk were defined as 1.5 for heterozygous and 2.25 for homozygous loci. The minor alleles were set up as the risk ones. The sizes of samples thus generated were the same as for the real data analyzed in this investigation, i.e., 652 and 4929 for controls and cases, respectively. The reference and simulated datasets were characterized by MAFs and decay of LD. The MAFs were counted with Plink 1.9 (--freq argument) and the decay of LD was estimated as the median values of r^2^ in 5000 bp bins with the custom script processing LD matrices.

## 5. Conclusions

Our research demonstrated that already existing genome-wide genotype data are valuable sources, and through using these data, new candidate loci can be identified with new approaches. Enlarging the multi-loci approaches in GWASs, which are believed to have more statistical power than the single-locus approaches, we proposed, validated and applied to real data a new workflow based on the clustering of SNPs. It exploited the unique features of the density-based spatial clustering algorithms DBSCAN and HDBSCAN. They can build clusters of any shape and do not require prior knowledge of the number of clusters. This allowed the grouping of the SNPs that are not necessarily consecutive, as did the previous approaches to partitioning the genome. We believe this model describes the genetic architecture better since it accounts for long-distance interactions between SNPs. The proposed workflow demonstrates the efficacy of DBSCAN and especially HDBSCAN for identifying genetic markers associated with disease. Both known and new loci associated with IS were revealed using the method proposed. In particular, one should emphasize the protocadherins, whose complex role in IS was first proposed based on genome-wide genotype data. The effective clustering of SNPS by DBSCAN and HDBSCAN provides potential solutions for several other problems, among which are the selection of tag SNPs and quality control in associative studies, as well as the “big-p, small-n” problem in the application of machine learning to genotypic data.

## Figures and Tables

**Figure 1 ijms-24-15355-f001:**
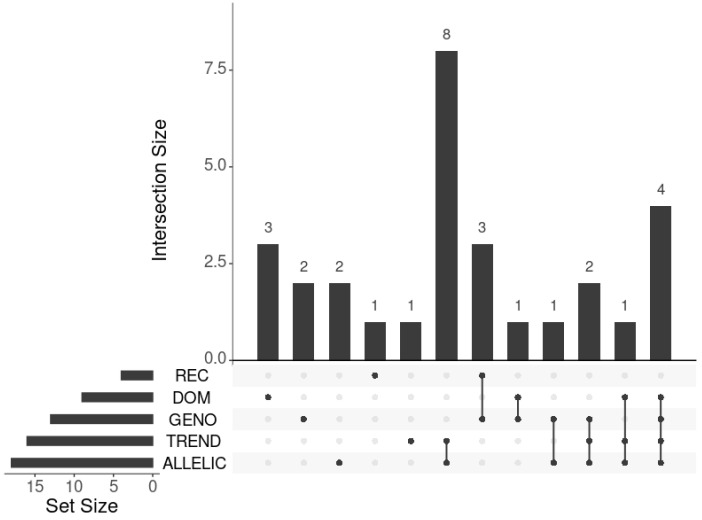
UpSet plot of significant SNPs obtained for different models under the traditional GWAS.

**Figure 2 ijms-24-15355-f002:**
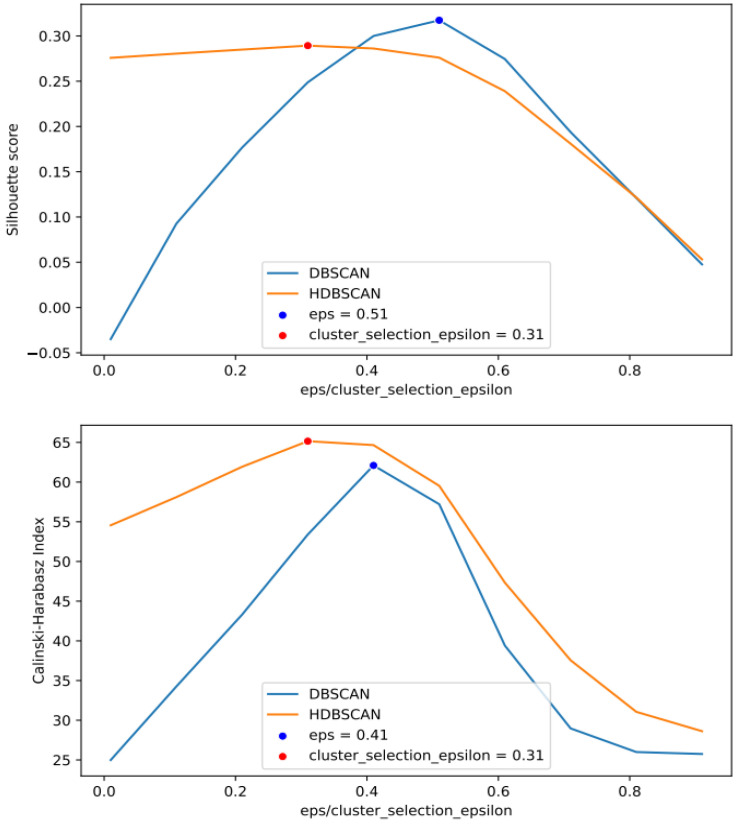
The changes in the quality of clusterization by arguments of clusterization functions for chromosome 1.

**Figure 3 ijms-24-15355-f003:**
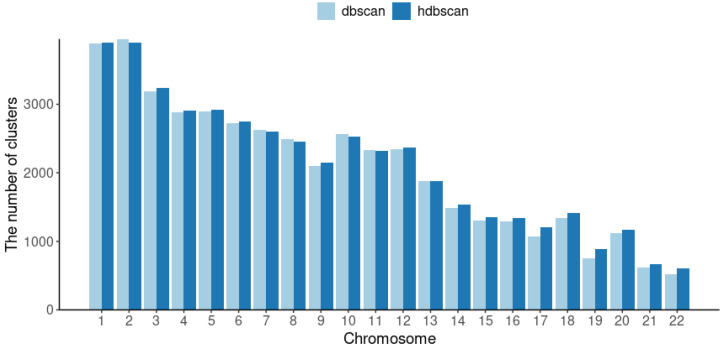
The number of clusters per chromosome obtained by DBSCAN and HDBSCAN.

**Figure 4 ijms-24-15355-f004:**
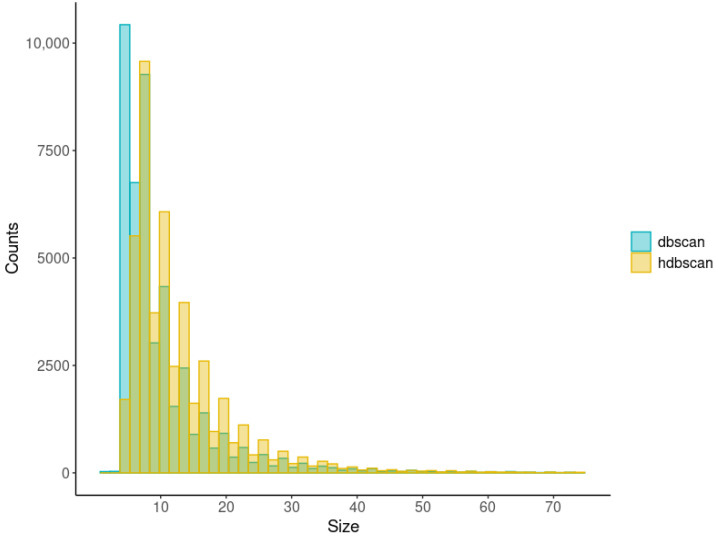
The histograms of cluster sizes (the *y*-axis was limited to the value of 75).

**Figure 5 ijms-24-15355-f005:**
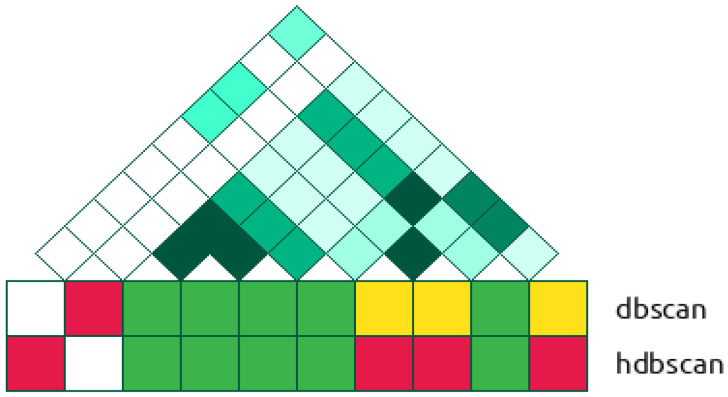
The composition of clusters formed and the heatmap of LD at the genomic region chr17:31860912-31881235 containing significantly associated LD-blocks of four SNPs (red color) identified with the HDSCAN algorithm. The tracks DBSCAN and HDBSCAN show the cluster memberships of SNPs. The SNPs from the same cluster within the track are marked by similar colors. The heatmap of LD represents r^2^ values (deep green denotes 1.0 and the white color denotes 0).

**Figure 6 ijms-24-15355-f006:**
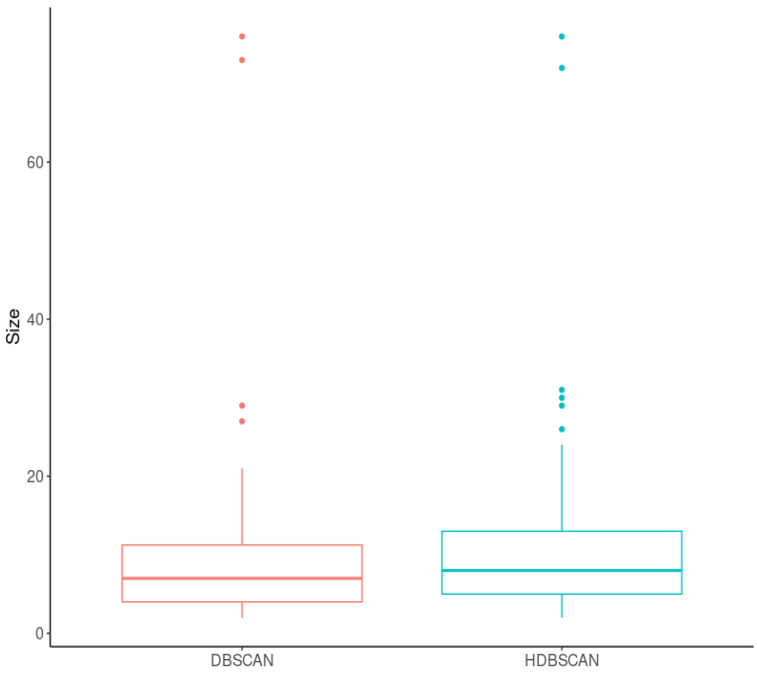
The distribution of sizes of the LD-blocks significantly associated with IS.

**Figure 7 ijms-24-15355-f007:**
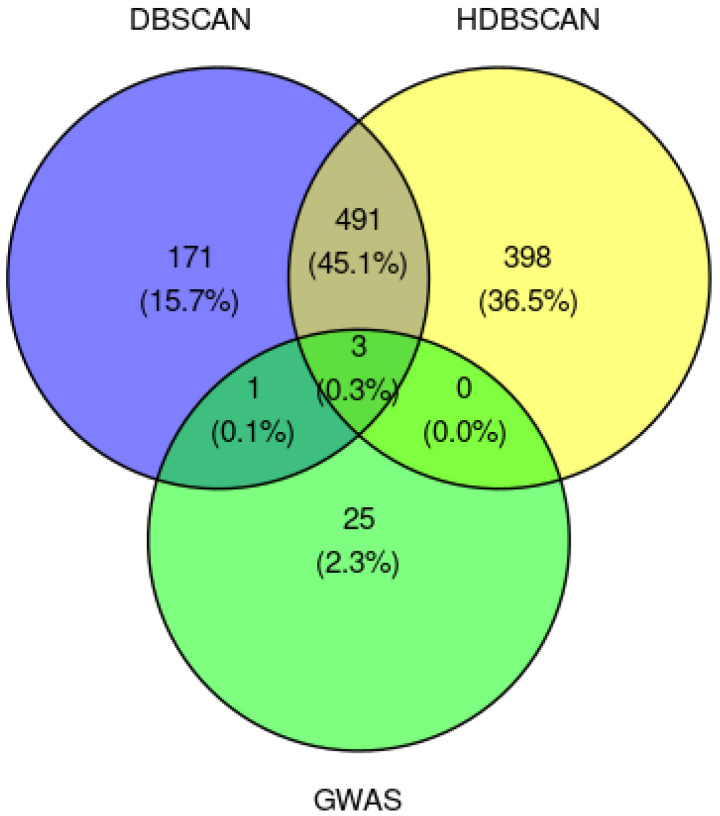
Venn diagram illustrating the overlap of the three approaches with each other. The number of risk polymorphisms obtained by classic GWAS and cluster-based approaches (DBSCAN and HDBSCAN) are depicted.

**Table 1 ijms-24-15355-t001:** Frequencies of SO terms.

Selected SNPs		All SNPs		SO Term
Freq	Prop, %	Freq	Prop, %	
1	3.1	12,278	1.2	3_prime_UTR_variant
1	3.1	37,018	3.7	downstream_gene_variant
16	50	377,372	38	intergenic_region
3	9.4	185,166	18.7	intragenic_variant
9	28.1	346,775	34.9	intron_variant
2	6.3	34,188	3.4	upstream_gene_variant

**Table 2 ijms-24-15355-t002:** The descriptive statistics of analyzed objects.

HDBSCAN	DBSCAN	Parameter
605,016	475,445	The number of polymorphisms clusterized
266,609	396,180	The number of polymorphisms not clusterized (noise)
0.69	0.55	The efficiency of clusterization
46,120	45,388	The number of clusters
5–384	1–385	The minimum and maximum size of clusters
10	8	The median of cluster sizes
0.29	0.27	The mean of Silhouette coefficient
56.4	53.2	The median of Calinski and Harabasz score
44,842	41,132	The number of LD-blocks
79	68	The number of LD-blocks significantly associated with IS
8	7	The median of sizes of LD-blocks associated with IS
892	666	The number of polymorphisms within LD-blocks associated with IS
122	98	The number of genes and loci associated with IS

**Table 3 ijms-24-15355-t003:** The common genes for our study and associated with IS from online projects.

Cluster-Based GWAS	Classic GWAS	Project
*RUNX1*, *LHFPL3*	*RUNX1*	GWAS Central
*USF1*, *CD34*, *KIF26B*, *RUNX1*	*RUNX1*	DisGeNET
–	*GPR26*	IDG
*KIF26B*, *LHFPL3*, *RUNX1*	*RUNX1*	Monarch Initiative

**Table 4 ijms-24-15355-t004:** Sizes of gene sets downloaded from public resources and available for examination in our research.

Tested	Downloaded	Project
351	400	GWAS Central
1037	1159	DisGeNET
278	323	IDG
116	131	Monarch Initiative

## Data Availability

The haplotypes of the 1000 Genomes project under the release Phase I were downloaded from https://ftp.1000genomes.ebi.ac.uk/vol1/ftp/phase1/analysis_results/supporting/omni_haplotypes (accessed on 1 September 2023). The corresponding recombination maps were downloaded from https://ftp.1000genomes.ebi.ac.uk/vol1/ftp/technical/working/20130507_omni_recombination_rates (accessed on 1 September 2023). The synthetic data created and used in this investigation are available in Plink binary format from the GitHub repository https://github.com/inzilico/synthetic-data.git (accessed on 10 September 2023). The genotype–phenotype data of IS were obtained from the repository of phenotypes and genotypes dbGaP and are available under the accession number phs000615.v1.p1 once the data access request is approved via the Authorized Access System of the repository.

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
