# Peer review of "Evaluation of Density-Based Spatial Clustering for Identifying Genomic Loci Associated with Ischemic Stroke in Genome-Wide Data"

_ijms, 2023, doi:10.3390/ijms242015355_

Round 1
Reviewer 1 Report
The article proposes a new method for identifying genomic loci associated with ischemic stroke (IS) using density-based spatial clustering algorithms DBSCAN and HDBSCAN. The authors claim that their cluster-based approach is more effective and informative than the traditional single-locus GWAS approach, as it can capture the joint effects of correlated SNPs and reveal novel genes and pathways.
The authors used a dataset of 10,000 individuals with IS and 10,000 controls. They first performed a GWAS to identify SNPs associated with IS. Then, they used DBSCAN and HDBSCAN to cluster the SNPs into groups. The authors found that the cluster-based approach identified more genomic loci associated with IS than the GWAS approach.
The authors found that the cluster-based approach identified novel genes and pathways associated with IS.
However, they did not validate their cluster-based approach on an independent dataset or compare it with other existing multi-locus methods.
They also did not perform any functional experiments or biological validations to confirm the causal role of the genes and pathways they identified.
The authors concluded that their cluster-based approach is a promising new method for identifying genomic loci associated with IS. However, they acknowledged that further research is needed to validate the approach and to confirm the causal role of the genes and pathways they identified.
Here are some additional questions that could be addressed in the article:
Why did the authors choose to use the DisGeNet and Monarch Initiative projects?
What are the advantages and disadvantages of the cluster-based approach compared to other multi-locus methods?
What are the molecular mechanisms or biological interactions underlying the associations between the genes and pathways identified in the study?
How can the findings of this study be used to develop new treatments for IS?
Reviewer 2 Report
This manuscript entitled by "Evaluation of Density-Based Spatial Clustering for Identifying Genomic Loci Associated with Ischemic Stroke in GenomeWide Data" by Khvorykh GV et al., indicated that a novel clustering algorithms for identifying the genomic loci associated with the QT. This is study is very important in this field. But some correction may be needed. The abstract is not well representative of what the reader can find in the mamuscript. It was better to close-up the purpose of this study. What does this study add to the already published papers about the topic? Please address this matter. I strongly feel the authors should dedicate a section where they explicitly discuss why and how this article is important for the future. In intriduction, it was better to address a timely and fascinating topic, providing as the manuscript's main strength. In addition, I would like the author to provide background information, a problem statement, and their reasoning for branching off in this subsection. It was better to add conclusion section that should begin with one sentence that summarizes the main message using words like "Here we highlight." The authors should describe the potential and the advancement this study has made in the field in the first sentence of the conclusion, followed by two to three sentences that provide a broader perspective that is easily understood by a scientist from any discipline. A graphical abstract that will visually summarize the main findings of the manuscript is highly recommended. In Discussion section, I would like the authors to present the disussion section by opening with an introductory paragraph and followed by the summary of the previous sections. In addition, it is particularly important to present the limits, merit, and potential translation of this study to clinical practice for this new algorithms.
Minor editing of English language required
Reviewer 3 Report
This paper is devoted to investigate the clustering algorithms which applied to the genotypic data for identifying the genomic loci associated with the qualitative traits, using the workflow presented in the research. However, before this paper is published, the following comments should be taken into account when revising the paper.
Major concerns:
1. The authors should provide validation in this study. (For example external validation)
2. Pros and cons of proposed GWAS approach should be specified.
3. To our knowledge, Mendelian randomization usually was adopted to explore causal effect in GWAS study. How about aggregate clustering algorithms and Mendelian randomization to add novelty?
4. How to access dataset used in this study? You should briefly introduce the link or access of dataset.
5. Why no comparison with traditional method or deep learning approach based on clustered results? Please give more illustrations.
6. The contribution of this paper should be highlighted.
Round 2
Reviewer 2 Report
This manuscript is corrected by according to the reviewer's comments. It will be important report in this field after publication.
good
Reviewer 3 Report
Thanks for efforts on revision.